# Experiences, distress and burden among neurologists in Norway during the COVID-19 pandemic

**Espen Saxhaug Kristoffersen**[1,2]*, **Bendik Slagsvold Winsvold**[3,4], **Else Charlotte Sandset**[4,5], **Anette Margrethe Storstein**[6], **Kashif Waqar Faiz**[1]

**1** Department of Neurology, Akershus University Hospital, Lørenskog, Norway, **2** Department of General Practice, University of Oslo, Oslo, Norway, **3** Department of Research, Innovation and Education, Division of Clinical Neuroscience, Oslo University Hospital, Oslo, Norway, **4** Department of Neurology, Oslo University Hospital, Oslo, Norway, **5** The Norwegian Air Ambulance Foundation, Oslo, Norway, **6** Department of Neurology, Haukeland University Hospital, Bergen, Norway

* e.s.kristoffersen@medisin.uio.no

## Abstract

### Background

The ongoing COVID-19 pandemic has caused rapid changes in the healthcare system. Workforce reorganization, reduced standard of care and a lack of personal protection equipment (PPE) for health care workers were among the concerns raised in the first wave of the pandemic. Our aim was to explore the experiences, distress and burden among Norwegian neurologists during the first weeks of the pandemic.

### Methods

Hospital-based neurologists in Norway (n = 400) were invited to a web-based survey in April 2020. The study focused on patient management, organizational changes and personal stress during the first weeks of the pandemic lockdown. Work-home interface stress was assessed by the Cooper Job Stress Questionnaire.

### Results

In total, 135 neurologists participated. Seventy-three% experienced a change in their personal work situation, and 67% examined patients with suspected COVID-19 infection and neurological disease. Changed access to resources, and the perception that medical follow-up was unsatisfactory, were associated with a high degree of burden and stress. Neurologists were also worried about the potential lack of PPE and the fear of spreading SARS CoV-2 to close family members. The mean score of work-home interface stress was 2.8 with no significant differences between gender or specialist status. Reduced standard of care was reported for all neurological conditions, and in particular for non-emergency treatments.

**Data Availability Statement:** All relevant data are within the manuscript and its Supporting information files.

**Funding:** The author(s) received no specific funding for this work.

**Competing interests:** BWS has received speaking fees from Novartis, unrelated to the present work. ECS has received speaking fees from Bayer and Novartis, unrelated to the present work. AMS,ESK, KWF and SHJ report no conflicts of interest. This does not alter our adherence to PLOS ONE policies on sharing data and materials.

## Conclusion

The vast majority of neurologists in Norway experienced a change in their personal work situation during the first phase of the pandemic. The fear of becoming infected and ill was not a major contributor to burden and stress.

## Introduction

The coronavirus disease of 2019 (COVID-19) was declared a pandemic by the World Health Organization in March 2020 and led to challenges in the delivery of medical care worldwide [1, 2]. Initially, there was much uncertainty, and different countries and regions chose different approaches. The major common goal was to reduce cross-infection with Severe Acute Respiratory Syndrome CoronaVirus 2 (*SARS-CoV-2*) and thus to protect patients and hospital staff in order to avoid critically overloading the healthcare system [3]. Workforce reorganization, changing and increased shift load and downscaling of usual care were among the initial efforts made [3]. In many countries this led to a reduced availability of in-hospital appointments and delayed treatment for chronic diseases [2]. A rapid shift in favor of telemedicine provided continuous access to care despite infection control measures for some patients, but for others this resulted in suboptimal consultations without the possibility of a proper clinical examination and treatment [4, 5]. While efforts were made on all levels of the healthcare system, physicians-including neurologists—remained on the frontline. Early reports of lack of personal protective equipment (PPE) and deaths among exposed physicians contributed to uncertainty and fear [6, 7].

During these first hectic weeks of the pandemic, it was reported that a proportion of COVID-19 patients presented with neurological symptoms [8, 9]. No consensus statements were in place for the management of neurological disorders in the very beginning of the pandemic. Thus, neurologists needed to handle uncertainty, potential risk of infection, and balancing the risk of being infected with the need to protect patients with chronic neurological disorders that were at increased risk if infected by COVID-19. In addition, workforce reorganization, long shifts, increased workload, physical exhaustion, social isolation, quarantine, inadequate PPE, uncertainty about virus transmission routes, and the need to make treatment decisions with reduced standard of care may have affected both physical and mental health [10]. The work-home interface stress may have increased as a consequence of work-related changes, but also out of fear to bring infection home to the family. The effect of this additional stress is of interest, in particular in light of the high rates of burnout among neurologists compared to other physicians and the general population [11, 12]. Such stressors may increase the risk of medical errors and reduced patient safety, which themselves are a risk factor for physician burnouts [12].

The objectives of this "Neurology during a pandemic" (NeuroPan) study were two-fold: i) to investigate the initial experiences of management of neurological patients, and ii) to investigate how the pandemic affected neurologists on a personal level.

## Materials and methods

### Design and setting

Norway has a population of approximately 5.4 million inhabitants, and 17 hospitals have a neurological department, varying from smaller district hospitals to larger university hospitals,

employing about 400 neurologists (approximately 50% residents and 50% senior consultants). Norwegian hospitals are almost exclusively publicly financed. Patients need a referral from a general practitioner in order to visit the hospital, except in emergencies. Norway has an all-covering national health insurance. Thus, all patients are referred on the same conditions and with the same threshold for further investigations, treatments and follow-ups. The first Norwegian national lockdown due to the COVID-19 pandemic was declared by March 12[th], 2020 and the schools and universities gradually re-opened from April 27[th], 2020.

The study was conducted as an anonymous online questionnaire survey among Norwegian neurologists about neurological diseases during the primary stage of the COVID-19 pandemic. The survey was distributed by e-mail in April 2020 to hospital-based Norwegian neurologists. The neurologists had approximately three weeks to answer the questionnaire, and the study were concluded in May 2020.

## Questionnaire and outcomes

The questionnaire was based on the authors' clinical experiences from the first weeks of the pandemic in addition to their general knowledge and experience in neurology and health services research. The questionnaire (S1 File) included background variables including age, sex, training status (resident/ senior consultant) and type of hospital (university hospital/ non-university hospital). Furthermore, the participants answered various questions regarding their personal considerations and their own handling of patients with neurological disorders during the pandemic. There were both general questions and questions specific for certain neurological conditions. All questions were asked in relation to the first weeks of the pandemic lockdown.

Perceived work-home interface stress was measured using a modified and previously validated version of the Cooper Job Stress Questionnaire [13, 14].

To the question "To what extent are the following situations/factors burdening (stressful) for you?", three statements were provided: "The job has a negative effect on my family life / on striking a balance between work and private life / The job has a negative effect on my social life". The statements were scored from 1 (no burden) to 5 (very high burden). The average scores for the three dimensions were calculated into a work-home interface stress score [15].

## Statistical analyses

For descriptive data, proportions, means, and standard deviations (SD) or 95% confidence intervals (CI) are given. The total number of responders to each question vary, as all follow-up questions were not relevant for all participants, e.g. a neurologist who did not do teleconsultations or treated headache patients during the pandemic did not receive further follow-up questions on this topic. These missing data points were handled by reducing the total number of respondents as applicable. Groups were compared using the *t*-test (continuous data) or the $\chi^2$ test (categorical data). Non-parametric tests were used as appropriate.

Statistical significance was defined by p<0.05, using a two-sided test. Statistical analyses were performed using IBM SPSS Statistics for Windows, Version 26.00 (SPSS Inc., Chicago, IL, USA).

## Ethics

In accordance to the Norwegian law on medical research, the project did not require an approval from the Regional Committee for Medical Research Ethics. The Data Protection Officer at Akershus University Hospital approved the study. Informed consent was obtained by agreeing to participate. All responses were collected anonymously with no identifiable

information gathered on the respondents by the study team. The NorPan study is registered in the COVID-19 trial registration at the Norwegian Clinical Research Infrastructure Network.

## Results

### Sample

In total, 135 neurologists answered the questionnaire. Of these, 58% (n = 78) were women, mean age was 42 years (SD 10.0, range 27–60), 58% (n = 78) were consultants and 60% (n = 81) worked at a university hospital (Table 1).

### Overall management at neurological departments

Overall, 73% reported a change in their personal work situation and 50% had their work schedule changed (Table 2). More residents than senior consultants had their work schedules changed (60% vs. 42%, p = 0.044) or had extended working hours (25% vs. 9%, p = 0.021). Sixty-seven % had assessed patients with clinical suspicion of COVID-19, with a significant higher proportion among residents than among senior consultants (77% vs. 60%, p = 0.038). There were no significant differences between residents and senior consultants or between women and men concerning work satisfaction during the lockdown.

Major changes in the clinical practice were reported (Tables 3 and 4).

**Table 1. Description of the sample (n = 135).**

|  | Men N = 57 | Women N = 78 | Residents N = 57 | Senior consultants N = 78 | Total N = 135 |
|---|---|---|---|---|---|
| Age, mean (SD) | 44.3 (11.7) | 40.3 (8.2) | 34.2 (4.2) | 47.8 (9.0) | 42.1 (10.0) |
| Gender, n (%) |  |  |  |  |  |
| Men | 57 (100) | 0 (0) | 25 (44) | 32 (41) | 57 (42) |
| Women | 0 (0) | 78 (100) | 32 (56) | 46 (59) | 78 (58) |
| Neurologist status, n (%) |  |  |  |  |  |
| Residents | 25 (44) | 32 (41) | 57 (100) | 0 (0) | 57 (42) |
| Senior consultants | 32 (56) | 46 (59) | 78 (0) | 78 (100) | 78 (58) |
| Type of hospital, n (%) |  |  |  |  |  |
| University hospital | 38 (67) | 43 (55) | 40 (70) | 41 (53) | 81 (60) |
| Non-university hospital | 19 (33) | 35 45) | 17 (30) | 37 (47) | 54 (40) |

**Table 2. What is your work situation like now as compared to normal? n (%).**

|  | Residents | Senior consultants | Total |
|---|---|---|---|
| Changed | 42 (76) | 52 (71) | 94 (73) |
| My duties are unchanged, but I work more than before | 8 (15) | 7 (11) | 15 (13) |
| My duties are changed, and I work more than before | 7 (13) | 16 (24) | 23 (19) |
| My duties are unchanged, but I work less than before | 17 (30) | 24 (36) | 41 (33) |
| I have been relocated from neurology to another ward | 3 (5) | 3 (4) | 6 (5) |
| Our work schedule has changed | 34 (60) | 31 (42) | 65 (50) |
| We have extended the doctors' working hours | 14 (25) | 7 (9) | 21 (16) |
| If you have a research position, has research time been revoked? | 7 (37) | 9 (22) | 16 (27) |
| We have been able to facilitate home office | 33 (63) | 46 (73) | 79 (69) |
| I am satisfied with the work | 47 (82) | 55 (73) | 102 (78) |
| Have you assessed patients with clinical suspicion of COVID-19? | 44 (77) | 47 (60) | 91 (67) |

**Table 3. Please rate the following statements, n (%).**

| | Strongly agree | Agree | Neither agree nor disagree | Disagree | Strongly disagree |
|---|---|---|---|---|---|
| Fewer patients have been referred to the emergency ward with potentially neurological conditions | 54 (40) | 61 (45) | 13 (10) | 6 (4) | 1 (1) |
| Fewer patients have been admitted from the emergency ward to the neurology department with potentially neurological conditions | 49 (36) | 59 (44) | 17 (13) | 9 (7) | 1 (1) |
| Patients who would normally be admitted to the neurology department for acute treatment are being sent home without admission | 3 (2) | 18 (13) | 38 (28) | 55 (41) | 20 (15) |
| Patients who would normally be admitted to the neurology department for further sub-acute investigations are being sent home without admission (investigations postponed) | 3 (2) | 33 (25) | 28 (21) | 52 (39) | 17 (13) |
| In-patient elective treatments are postponed | 20 (15) | 72 (54) | 22 (16) | 17 (13) | 3 (2) |
| In-patient elective investigations are postponed | 22 (16) | 80 (60) | 20 (15) | 11 (8) | 1 (1) |
| Patients have come for treatment and admissions later than usual (patient delay) | 17 (13) | 78 (59) | 31 (23) | 6 (5) | 1 (1) |
| Hospitalised patients are considerably sicker now than before | 7 (5) | 37 (28) | 53 (40) | 32 (24) | 5 (4) |
| Reduced standard of care is available for patients with acute neurological conditions | 7 (5) | 25 (19) | 27 (20) | 56 (42) | 19 (14) |
| Reduced standard of care is available for patients with chronic neurological conditions | 24 (18) | 62 (46) | 30 (22) | 16 (12) | 3 (2) |
| Patients with acute neurological conditions had a worse prognosis | 3 (2) | 15 (11) | 35 (26) | 55 (41) | 25 (19) |
| Patients with chronic neurological conditions had a worse prognosis | 7 (5) | 19 (14) | 63 (47) | 39 (29) | 6 (5) |
| I experienced that the academic community in Norway collaborated to find good solutions for neurological patients | 19 (14) | 64 (48) | 40 (30) | 9 (7) | 2 (2) |
| I experienced that the health authorities in Norway collaborated to find good solutions for neurological patients | 7 (5) | 34 (25) | 73 (54) | 18 (13) | 3 (2) |

Fifty-three % reported that the number of neurological in-patient beds was reduced. The majority (85%) agreed that fewer patients had been referred to the emergency ward with neurological conditions during the pandemic lockdown. Further, 80% agreed that fewer patients than usual had been admitted from the emergency ward to the neurological ward. Patient delay was common (72%). When patients with acute neurological disorders were admitted, most were treated as usual, but in-patient planned evaluations and treatments were postponed. Only 8% maintained a regular out-patient clinic with in-person appointments. Eighty-seven % reported a shift towards more telemedicine, with significantly more use of telephone than video consultations for both newly referred patients (54% vs. 30%, p<0.001) and follow-ups (99% vs. 50%, p<0.001).

Only 30% agreed that the healthcare authorities in Norway collaborated to find good solutions for neurological patients, but 62% agreed that the academic community collaborated on these topics (Table 3). Twenty-four % agreed that the standard of care was reduced for patients with acute neurological conditions whereas 64% agreed that the standard of care was reduced for patients with chronic neurological conditions (Table 3).

## Management of different neurological disorders

Reduced standard of care was reported by 48% for stroke, 30% for epilepsy, 42% for headache, 71% for multiple sclerosis, 74% of movement disorders, 56% of amyotrophic lateral sclerosis (ALS), 20% for glioblastoma, 15% for inflammatory or immunological neuropathies and 23%

**Table 4. Management of ten different neurological disorders during the initial phase of the COVID-19 pandemic, n (%).**

|  | n (%) |
|---|---|
| **Stroke** |  |
| The stroke care is changed | 45 (45) |
| We have changed the thrombolysis procedures | 80 (83) |
| We are still admitting patients with TIA | 86 (90) |
| Fewer stroke patients are coming to the hospital | 90 (89) |
| Out-patients clinic follow-ups are postponed | 47 (57) |
| Stroke patients have reduced rehabilitation options | 58 (64) |
| **Epilepsy** |  |
| We have maintained the regular epilepsy care | 51 (51) |
| Acute treatment for admitted patients with seizures is changed | 9 (10) |
| Status epilepticus treatment is worse than usual | 4 (4) |
| As many patients as usual are sedated and intubated as part of status epilepticus treatment | 63 (89) |
| There is worse access to emergency EEG | 16 (17) |
| Out-patient follow-up are primarily conducted by telephone consultation | 91 (95) |
| Out-patient follow-ups are postponed | 57 (58) |
| Access to elective EEG and sleep-deprived EEG is worse | 48 (53) |
| We take fewer medication analysis | 25 (27) |
| **Headache** |  |
| We have maintained the regular headache care | 39 (52) |
| We continue to offer botulinum toxin at regular intervals | 31 (43) |
| Have you switched more patients than usual from botulinum toxin to CGRP antibodies? | 11 (20) |
| Have you been more likely to put patients on CGRP antibodies rather than botulinum toxin? | 27 (43) |
| **Multiple sclerosis** |  |
| We have maintained the regular multiple sclerosis care | 11 (18) |
| Treatment of acute attacks is less available | 12 (20) |
| I have asked patients to stay away from the hospital because they are a vulnerable group | 37 (63) |
| We have chosen different types of treatment for newly diagnosed patients than we normally do | 35 (65) |
| We have changed the current treatment for individual patients due to the pandemic | 12 (22) |
| Out-patient follow-up are primarily conducted by telephone consultation | 54 (93) |

(*Continued*)

**Table 4.** (Continued)

|  | n (%) |
|---|---|
| Out-patient follow-ups are postponed | 33 (56) |
| I have read the recommendations from the National Advisory Unit on multiple sclerosis and Covid-19 | 56 (92) |
| **Movement disorders** |  |
| We have maintained the regular movement disorders care | 15 (28) |
| Out-patient follow-up are primarily conducted by telephone consultation | 39 (82) |
| Out-patient follow-ups are postponed | 39 (74) |
| I have patients whose advanced Parkinson's treatment has been postponed as a consequence of the pandemic | 12 (28) |
| **Amyotrophic lateral sclerosis** |  |
| We have maintained regular ALS out-patient clinic | 9 (60) |
| We offer telephone consultations to ALS patients | 13 (81) |
| I have recommended that ALS patients avoid hospital because they should not be exposed to potential Covid-19 infection at the hospital | 13 (81) |
| I have patients whose home respirator has been delayed | 1 (7) |
| I have patients who has been denied home respirator | 0 (0) |
| Patients with ALS have a decreased lifespan due to worse options from the health services during the pandemic | 2 (13) |
| **Glioblastoma** |  |
| Out-patient follow-ups are postponed | 7 (32) |
| Out-patient follow-ups are postponed because these patients should not be exposed to potential Covid-19 infection at the hospital | 6 (29) |
| Treatment with temozolomide is given as planned | 18 (95) |
| Diagnostic examination of glioblastoma patients takes longer time | 0 (0) |
| There is longer waiting time before surgery for newly diagnosed glioblastoma patients | 0 (0) |
| Radiation therapy are postponed | 1 (7) |
| We avoid admitting patients with glioblastoma even when their condition is worsening because they should not be exposed to potential Covid-19 infection at the hospital | 1 (5) |
| Patients with glioblastoma have a decreased lifespan due to worse options from the health services during the pandemic | 1 (6) |
| **Immune-mediated polyneuropathies** |  |
| Patients receive immunological treatment with regular intervals | 23 (85) |
| Immunological treatment of new patients begins as planned | 24 (92) |
| We have postponed all treatment until we know more about Covid-19 and this type of treatment | 1 (4) |
| **Dystonia** |  |
| We continue to offer botulinum toxin at regular intervals | 21 (81) |
| **Spasticity** |  |
| We continue to offer botulinum toxin at regular intervals | 23 (85) |

for dystonia and spasticity. Table 4 shows a more detailed description of the reported management of ten different neurological disorders during the initial phase of the lockdown.

Overall, the acute management was mostly reported to be unchanged, while there were several changes to planned activities, in particular out-patient follow-ups which were postponed or conducted by teleconsultations.

Regarding stroke patients, 57% of the respondents reported that out-patient clinic follow-ups were postponed, and 64% reported that rehabilitation options were reduced.

Acute treatment of seizures, and EEG in the acute phase, were not strongly impacted by the pandemic. Access to elective EEG and sleep-deprived EEG was reduced and follow-ups were mainly conducted by telephone for epilepsy patients.

More than half (57%) reported that migraine patients were not treated by regular intervals with botulinum toxin A (BTX) injections. Almost half of the respondents (43%) were more likely to put patients on CGRP antibodies rather than BTX. However, BTX was offered at regular intervals to patients with dystonia or spasticity.

Only 18% reported that regular multiple sclerosis care was maintained. More specifically, 65% reported that they had chosen different types of treatment for newly diagnosed patients than they normally would do. Twenty-two % of the neurologists had changed the current treatment scheme for individual patients due to the pandemic. Telephone consultations were the main form of follow-up, and 63% had asked patients to avoid the hospital because they considered patients with multiple sclerosis a vulnerable group to COVID-19.

Management and care of movement disorders were changed in 72%, and 28% had patients whose advanced Parkinson's treatment was postponed.

Sixty % maintained regular ALS out-patient clinics, but 81% had recommended the ALS patients to avoid the hospital in order to reduce the risk of being exposed to COVID-19 infection. No neurologists reported that patients with ALS were denied home respirator treatment due to the pandemic.

Glioblastoma patients were treated as normal. Almost all reported that diagnostic examinations, treatment with temozolomide, surgery and radiation therapy were given as planned.

## Self-perceived sleep problems, depression and distress among neurologists

Thirteen % (n = 17) of the neurologists reported that their sleep had suffered due to the pandemic and 15% (n = 19) had felt depressed for more than 14 out of the last 28 days. There were no significant differences between residents and senior consultants, or between those who assessed and did not assess potential COVID-19 infected patients with regards to suffering from sleep problems and feeling depressed.

A high degree of burden and stress was associated with changed access to resources and that patients were not given the follow-up they should have received (Table 5).

Furthermore, perceived stress was associated with the potential lack of PPE and the fear of spreading SARS CoV-2 to close family members. The danger of becoming infected and ill was not perceived as an important contributor to burden and stress, and almost 80% were satisfied with their work situation during the pandemic.

Neurologists with self-perceived depressed thoughts reported significantly higher burden/stress for the following statements: The fear of contracting SARS CoV-2 at work impacts negatively on my quality of life (p = 0.031), I am afraid that I will spread SARS CoV-2 to close family members (p = 0.001), Potential lack of personal protection equipment in my clinical work (p = 0.04), Patients are not given the follow-up they should receive (p = 0.032) and Changed work routines and access to resources (p = 0.001).

**Table 5. To what degree are you stressed or burdened by the following situations/factors related to the COVID-19 pandemic, n (%)? 1 = No burden and 5 = Very significant burden.**

| | 1 | 2 | 3 | 4 | 5 | Mean value (SD) | Median value |
|---|---|---|---|---|---|---|---|
| My job impacts negatively on my family life | 29 (22) | 30 (22) | 41 (30) | 28 (21) | 7 (5) | 3.7 (1.2) | 3 |
| Finding balance between my work and private life | 25 (19) | 33 (24) | 32 (24) | 34 (25) | 11 (8) | 2.8 (1.2) | 3 |
| My job impacts negatively on my social life | 25 (19) | 31 (23) | 32 (24) | 35 (26) | 12 (9) | 2.8 (1.3) | 3 |
| The fear of contracting SARS CoV-2 at work impacts negatively on my work | 52 (39) | 48 (36) | 16 (12) | 15 (11) | 4 (3) | 2.0 (1.1) | 2 |
| The fear of contracting SARS CoV-2 at work impacts negatively on my quality of life | 59 (44) | 42 (31) | 19 (14) | 12 (9) | 3 (2) | 2.0 (1.1) | 2 |
| I am afraid that I will spread SARS CoV-2 to close family members | 24 (18) | 37 (27) | 35 (26) | 26 (19) | 13 (10) | 2.8 (1.2) | 3 |
| Potential lack of personal protection equipment in my clinical work | 31 (23) | 35 (26) | 36 (27) | 24 (18) | 9 (7) | 2.6 (1.2) | 3 |
| Patients are not given the follow-up they should receive | 7 (5) | 36 (27) | 54 (40) | 35 (26) | 3 (2) | 2.9 (0.9) | 3 |
| Changed work routines and access to resources | 16 (12) | 31 (23) | 45 (33) | 38 (28) | 5 (4) | 2.9 (1.1) | 3 |

The mean score (SD) of work-home interface stress was 2.8 (1.1) for the total sample with no significant differences between female (2.9 (1.1)) and male (2.6 (1.1)) or between residents (2.9 (1.0)) and senior consultants (2.6 (1.2)). Depressive thoughts were associated with significantly higher work-home interface stress (3.4 vs. 2.6, p = 0.005). Self-reported sleep problems were associated with significantly higher work-home interface stress (3.6 vs. 2.6, p = 0.001).

## Discussion

There were major changes in clinical neurological practice during the first weeks of the pandemic lockdown in Norway. Almost three out of four neurologists experienced a change in their personal work situation, and the majority examined patients with suspected COVID-19 infection and neurological disease. The pandemic affected various aspects of their personal and professional life. A high degree of burden and stress was associated with changed access to resources and to the fact that patients were not given necessary follow-up. Neurologists were also worried about the potential lack of PPE and the fear of spreading SARS CoV-2 to close family members. Interestingly, the danger of becoming infected and ill was not reported as an important contributor to burden and stress, and almost 80% were satisfied with their work overall situation during the pandemic.

Non-urgent follow-ups, such as out-patient clinic appointments were postponed or carried out as telephone consultations, and rehabilitation options were reduced. Delivering sub-optimal health care could impact on the short-term quality of life for patients with conditions such as headache and epilepsy, and on long-term outcome in other neurological conditions, e.g. stroke or multiple sclerosis. On the other hand, necessary urgent treatment of serious conditions, such as acute stroke, acute seizures, glioblastoma and immune-mediated polyneuropathies, was reported to be virtually unchanged during the first weeks of the pandemic. In Norway, as in Germany, a decrease in the absolute thrombolysis treatment numbers has been reported, but with similar treatment rates compared with previous months for those with ischemic stroke that presented to hospital within the thrombolysis time window [16–18].

The shift towards more teleconsultations is similar to what has been reported elsewhere [19–21]. Some neurological disorders may be better suited for telemedicine than others, and overall, this shift may be less problematic within non-emergency neurology than it is in other medical fields [22–28]. Many neurological patients have increased risk for severe COVID-19 due to their age, comorbidities or conditions that require immunosuppressive treatments. A large proportion of the neurologists in the present study had asked patients with multiple sclerosis, glioblastoma and ALS to stay home and postpone follow-ups in order to avoid being

infected with COVID-19 at the hospital. For these patients, teleconsultations may provide a safe way to maintain follow-up care.

Some academic and scientific communities were making international protocols and consensus statements rapidly available while others used more time. A recent consensus statement on good clinical practice regarding patients with neurological disease during the COVID-19 pandemic will hopefully provide guidance for neurologists and be an important contribution in the second wave of the pandemic [29].

The global community was clearly not sufficiently prepared for such a pandemic [30–33]. In most places it was local initiatives and heroic efforts from health care workers, more than governmental plans, that were responsible for the rapidly implemented changes needed to treat a high number of COVID-19 patients [3]. These fast re-organizations show that it is possible to implement fast-track changes if deemed necessary from a professional point of view. However, unplanned, fast and large scale changes may come with a cost for both patients and health care workers [10]. Only 30% of the respondents agreed that the health authorities in Norway collaborated to find good solutions for neurological patients, but 62% agreed that the academic community collaborated on these topics. This is in line with findings from other countries, where a lack of coordination and preparedness has been found and criticised [30, 31, 33].

It may be considered a paradox that 78% of the respondents reported that they were satisfied with the overall work situation in a period with much uncertainty and rapid shifts in availability of resources. One explanation may be that healthcare workers find it meaningful to contribute in the time of crisis, and may find the situation more satisfying than usual care despite relocations to other wards, downsized ward teams, less rest between shifts, longer shifts and more patients per shift.

The Norwegian neurologists experienced stress from the potential lack of PPE and the fear of spreading SARS CoV-2 to patients or close family members. The Norwegian government expressed early a shortage in PPE and required health care workers to reuse PPE. Furthermore, health care workers were told to use PPE only in cases of a high suspicion of COVID-19 despite widespread community spread, uncertainty about air transmission and concerns about asymptomatic transmission [34, 35]. A recent study among US neurologists found that during the initial weeks of the pandemic, most institutions had expressed shortages in PPE, with almost half (45%) of neurologists having to reuse their PPE [21].

Interestingly, the thought of becoming infected and ill was not associated with a substantial burden and stress among Norwegian neurologists, even though media reports from other countries suggested a high incidence, and even mortality for doctors on the frontline [36]. A recent meta-analysis found pooled prevalence of 23, 23 and 34% for anxiety, depression and insomnia among health care workers during the initial phase of the COVID-19 pandemic [10].

A study of frontline doctors in Pakistan reported a 43% prevalence of anxiety/depression in the initial weeks of the pandemic [37]. Further, a majority of the physicians dealing with suspected or confirmed COVID-19 cases in an Iraqi Kurdistan study reported high levels of insomnia and stress [38]. Two studies among health care workers in highly affected areas in Italy both reported high levels of burnout and psychological symptoms during the COVID-19 emergency, but with no significant differences between physicians and nurses [39, 40].

We found lower rates, with 13% reporting sleep problems and 15% reporting depressive thoughts due to the pandemic, but as different questions and comparisons were used in previous reports, these may not be directly comparable. The reasons for such high psychological distress may be diverse, but a study of Canadian emergency physicians reported that the following factors had an impact on physician well-being during the initial phase of the pandemic:

personal safety, academic and educational work, PPE, the workforce, patient volumes, work patterns, and work environment [41]. These reported factors are in line with our findings. Some of the stress related to an unknown and rapidly evolving situation is probably unavoidable. However, it may be hypothesized that an increased preparedness from the government could reduce the stress and burden placed on many health care workers during a pandemic with an unknown agent such as *SARS-CoV-2*. First of all, access to adequate PPE, but also realistic plans on how to strategically use the workforce including work schedules, academic and social support, and reasonable time to recover must be in place.

Interestingly, we found no significant differences in sleep problems and depressive thoughts between residents and senior consultants, or between those who assessed and did not assess potential COVID-19 infected patients. It may be hypothesised that residents on the frontline assessing potential COVID-19 patients have a higher chance of experiencing quarantine leading to isolation, loss of social support and negative thoughts. In addition, younger residents are more inexperienced, their educational activities halted, and they received reduced supervision from more senior physicians. In total, this could have led to more distress among residents than senior consultants. However, our findings do not point to any of these associations.

The combination of i) work related factors such as rapid and unpredictable changes in the work situation, the potential lack of PPE, the fear of spreading SARS CoV-2 to close family members, with ii) private life factors such as reduced contact with close family or caring for family members in distress, reduced social contact and suspension of personal travels may have led to work-home interface stress [15]. The mean score of work-home interface stress was 2.8 for the total sample with no significant differences between residents (2.9) and senior consultants (2.6). Neurologist with self-reported depressive thoughts or sleep problems had significantly higher work-home interface stress. The cross-sectional design in the present study does not permit any conclusions about causality or direction of the association between these outcomes. A study of Norwegian physicians from 2019 reported lower mean scores of 2.5 and 2.3, 4 and 15 years after graduation [42]. Based on this it is reasonable to assume that neurologists experienced slightly more work-home interface stress during the initial phase of the pandemic. However, this comparison should be made with caution as previous studies have shown that neurologists have a higher degree of stress, less job satisfaction and more burnouts than other physicians [11, 12]. Thus, our findings may reflect this previously described difference between neurologists and other physicians.

Respondents represented all neurological departments in Norway, which should ensure good representativeness. There are approximately 400 neurologists in Norway, of which 135 answered the questionnaire. However, a proportion of the non-responders will not have been in clinical work during the relevant period because of research or education terms, rotation to other wards or being in quarantine, on sick leave, or parental leave. Thus, we consider the 34% responder rate a conservative estimate. Given that all data were collected during three hectic and uncertain weeks when many changes were implemented for neurologists at the hospitals, we consider the responder rate satisfactory. Further, based on the age and gender diversity of the sample we do not suspect major biases in responder rate. Questionnaire studies such as this may introduce recall bias, however, we do not consider a reason to suspect systematic bias. The survey was sent out at the beginning of the pandemic in Norway when most of the hospitals made rapid changes in protocols and there was much uncertainty. Thus, the fact that the first wave of the pandemic ended up being better controlled in Norway than in many other countries may not have influenced the answers at this point, and we believe the results to some extent may be generalized to other countries with similar health care systems. As the world is facing the second wave of the pandemic, now with much more knowledge about COVID-19, it

would be interesting to repeat the questionnaire to describe changes in the delivery of health care and in the impact on neurologists.

## Conclusion

Three out of four neurologists in Norway experienced a change in their personal work situation during the first phase of the COVID-19-pandemic. Reduced standard of care was reported for all neurological conditions, particularly for non-emergency treatments. Changed access to resources, and the perception that medical follow-up was unsatisfactory, were associated with a high degree of burden and stress among neurologists. The fear of becoming infected and ill themselves was not reported as an important contributor to burden and stress.

## Supporting information

**S1 File. The complete questionnaire of NeuroPan.**
(PDF)

## Acknowledgments

The authors want to express their sincere gratitude to the participants and hospitals, without them the study would not have been possible.

## Author Contributions

**Conceptualization:** Espen Saxhaug Kristoffersen, Bendik Slagsvold Winsvold, Else Charlotte Sandset, Anette Margrethe Storstein, Kashif Waqar Faiz.

**Data curation:** Espen Saxhaug Kristoffersen.

**Formal analysis:** Espen Saxhaug Kristoffersen, Bendik Slagsvold Winsvold, Else Charlotte Sandset, Anette Margrethe Storstein, Kashif Waqar Faiz.

**Investigation:** Espen Saxhaug Kristoffersen, Bendik Slagsvold Winsvold, Else Charlotte Sandset, Anette Margrethe Storstein, Kashif Waqar Faiz.

**Methodology:** Espen Saxhaug Kristoffersen, Bendik Slagsvold Winsvold, Else Charlotte Sandset, Anette Margrethe Storstein, Kashif Waqar Faiz.

**Project administration:** Espen Saxhaug Kristoffersen, Bendik Slagsvold Winsvold, Else Charlotte Sandset, Anette Margrethe Storstein, Kashif Waqar Faiz.

**Writing – original draft:** Espen Saxhaug Kristoffersen.

**Writing – review & editing:** Espen Saxhaug Kristoffersen, Bendik Slagsvold Winsvold, Else Charlotte Sandset, Anette Margrethe Storstein, Kashif Waqar Faiz.

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
