## [Decision Letter · Decision Letter 0]

11 Jan 2021

PONE-D-20-40374

Experiences, distress and burden among neurologists during the COVID-19 pandemic

PLOS ONE

Dear Dr. Kristoffersen,

Thank you for submitting your manuscript to PLOS ONE. After careful consideration, we feel that it has merit but does not fully meet PLOS ONE’s publication criteria as it currently stands. Therefore, we invite you to submit a revised version of the manuscript that addresses the points raised during the review process.

Please provide minor revisions as suggested-

We look forward to receiving your revised manuscript.

Kind regards,

Emanuele Bartolini, MD

Academic Editor

PLOS ONE

Additional Editor Comments:

Dear Prof Saxhaug Kristoffersen

Your study is well-crafted and received favourable opinions from Reviewers.

Please fulfill their suggestions in order to further improve your paper quality

Journal Requirements:

2. In line with PLOS' guidelines on replicability and reporting (https://journals.plos.org/plosone/s/criteria-for-publication#loc-3), please include additional information regarding the survey or questionnaire used in the study and ensure that you have provided sufficient details that others could replicate the analyses. For instance, if you developed a questionnaire as part of this study and it is not under a copyright more restrictive than CC-BY, please include a copy, in both the original language and English, as Supporting Information.

"BWS has received speaking fees from Novartis, unrelated to the present work. ECS has received speaking fees from Bayer and Novartis, unrelated to the present work. AMS, ESK, KWF and SHJ report no conflicts of interest."

Reviewers' comments:

Reviewer's Responses to Questions

**Comments to the Author**

1. Is the manuscript technically sound, and do the data support the conclusions?

Reviewer #1: Yes

Reviewer #2: Yes

2. Has the statistical analysis been performed appropriately and rigorously? 

Reviewer #1: Yes

Reviewer #2: Yes

3. Have the authors made all data underlying the findings in their manuscript fully available?

Reviewer #1: Yes

Reviewer #2: Yes

4. Is the manuscript presented in an intelligible fashion and written in standard English?

Reviewer #1: Yes

Reviewer #2: Yes

5. Review Comments to the Author

Reviewer #1: This very informative and well written analysis (manuscript ID of PONE-D-20-40374) reports experiences, distress and burden among Norwegian neurologists during the first weeks of the COVID-19 pandemic based on a web-based survey. The authors report changes in the personal work situation for the vast majority of neurologists in Norway with a high degree of burden and distress during the early pandemic, which was mainly driven by changed access to resources, fearing lack of personal protection equipment and spreading the virus to family members rather than becoming ill. This work is an important contribution to understand the burden of the pandemic among physicians. The authors identify problems and fears of Norwegian neurologists that could be helpful for this still ongoing pandemic. I have only minor concerns regarding this manuscript. My detailed comments and questions are as follows:

Methods:

This descriptive analysis is well demonstrated. The data is fully available.

Am I right that the online questionnaire was send out and received again in April 2020? How much time did the participates have to consent to the study an answer the questionnaire? Recall bias is the main limitation, especially if the time window for answering this questionnaire was too small in this busy phase of the early pandemic.

Results/Discussion:

Was there an association between depressive thoughts and the factors that caused burden and stress to the neurologists, especially for those factors that caused a high degree of burden (changed access to resources, fearing lack of personal protection and spreading the virus to family members)?

It would be interesting to know the authors´ opinion about how to tackle some of the problems that caused stress during the early pandemic. Thus, it might be possible to reduce the burden of the still ongoing pandemic. For example, what do the authors think about screening health care workers for depressive symptoms in such an exceptional situation like a pandemic?

Reduced standard of care for stroke was reported by 48% of the participants and 83% reported changes in the thrombolysis procedures. This is very interesting. Are there data available concerning the percentage of treatment with thrombolysis in AIS patients in the early phase of the pandemic in Norway? For example, other countries recently reported a decrease in the absolute thrombolysis and thrombectomy treatment numbers but similar treatment rates compared with previous months for those AIS patients that presented to hospital (https://doi.org/10.1161/STROKEAHA.120.033160).

Reviewer #2: I read with much interest this paper. In general terms it is well written and clearly organised. I have just a few minor comments:

- I would recommend to specify exact time frame and place in the context of the pandemic wave in Norway providing details on the situation at that time (ie lockdown other measures). This because in 10 years time some people may not recollect many details.

- I would recommend citing other studies who have investigated the same issues in other countries or in other medical specialties and to discuss differences

- I would recommend to specify in the title that the paper focuses on neurologists in Norway

6. PLOS authors have the option to publish the peer review history of their article (what does this mean?). If published, this will include your full peer review and any attached files.

Reviewer #1: No

Reviewer #2: No

---

## [Author Response · Author response to Decision Letter 0]

20 Jan 2021

Please see attached file with the point by point responses to the reviewers

---

## [Editor Report · Decision Letter 1]

22 Jan 2021

Experiences, distress and burden among neurologists in Norway during the COVID-19 pandemic

PONE-D-20-40374R1

Dear Dr. Kristoffersen,

We’re pleased to inform you that your manuscript has been judged scientifically suitable for publication and will be formally accepted for publication once it meets all outstanding technical requirements.

Kind regards,

Emanuele Bartolini, MD

Academic Editor

PLOS ONE

---

## [Editor Report · Acceptance letter]

26 Jan 2021

PONE-D-20-40374R1 

Experiences, distress and burden among neurologists in Norway during the COVID-19 pandemic 

Dear Dr. Kristoffersen:

I'm pleased to inform you that your manuscript has been deemed suitable for publication in PLOS ONE. Congratulations! Your manuscript is now with our production department. 

Kind regards, 

on behalf of

Dr. Emanuele Bartolini 

Academic Editor

PLOS ONE